# Whole-exome sequencing of cell-free DNA and circulating tumor cells in multiple myeloma

S. Manier[1,2,3], J. Park[1,4], M. Capelletti[1], M. Bustoros [1], S.S. Freeman[5], G. Ha[5], J. Rhoades[5], C.J. Liu[1], D. Huynh[1], S.C. Reed[5], G. Gydush[5], K.Z. Salem[1], D. Rotem[5], C. Freymond[1], A. Yosef[1], A. Perilla-Glen[1], L. Garderet[6], E.M. Van Allen [1,5], S. Kumar [7], J.C. Love [5], G. Getz[5], V.A. Adalsteinsson[5] & I.M. Ghobrial[1,4,5]

Liquid biopsies including circulating tumor cells (CTCs) and cell-free DNA (cfDNA) have enabled minimally invasive characterization of many cancers, but are rarely analyzed together. Understanding the detectability and genomic concordance of CTCs and cfDNA may inform their use in guiding cancer precision medicine. Here, we report the detectability of cfDNA and CTCs in blood samples from 107 and 56 patients with multiple myeloma (MM), respectively. Using ultra-low pass whole-genome sequencing, we find both tumor fractions correlate with disease progression. Applying whole-exome sequencing (WES) to cfDNA, CTCs, and matched tumor biopsies, we find concordance in clonal somatic mutations (~99%) and copy number alterations (~81%) between liquid and tumor biopsies. Importantly, analyzing CTCs and cfDNA together enables cross-validation of mutations, uncovers mutations exclusive to either CTCs or cfDNA, and allows blood-based tumor profiling in a greater fraction of patients. Our study demonstrates the utility of analyzing both CTCs and cfDNA in MM.

---

[1] Medical Oncology, Dana-Farber Cancer Institute, Harvard Medical School, Boston, MA 02115, USA. [2] Hematology Department, CHU, Univ. Lille, 59000 Lille, France. [3] INSERM UMR-S1172, 59000 Lille, France. [4] Brigham and Women's Hospital, Boston, MA 02115, USA. [5] Cancer Program, Broad Institute of MIT and Harvard, Cambridge, MA 02142, USA. [6] Department of Hematology, St-Antoine University Hospital, Paris 75000, France. [7] Department of Hematology, Mayo Clinic, Rochester, MN 55902, USA. These authors contributed equally: S. Manier, J. Park. Correspondence and requests for materials should be addressed to V.A.A. (email: viktor@broadinstitute.org) or to I.M.G. (email: Irene_Ghobrial@DFCI.harvard.edu)

Multiple myeloma (MM) is a hematologic malignancy characterized by a bone marrow infiltration of clonal plasma cells with heterogeneous involvement in many areas of the bone marrow[1]. MM evolves from pre-malignant stages to symptomatic MM and remains incurable, due to intrinsic and acquired therapeutic resistance[1,2]. Better methods to track clonal evolution in MM may inform clinical management, and mounting evidence suggests that circulating tumor cells (CTCs) and cell-free DNA (cfDNA) may enable minimally invasive, genomic characterization of cancers including MM[3–5]. However, CTCs and cfDNA are rarely profiled together and thus it remains largely unknown whether CTCs and cfDNA reflect the same or different tumor clones and how these relate to a conventional tumor biopsy. We reasoned that ultra-low pass whole-genome sequencing (ULP-WGS) can provide a rapid and affordable screening and monitoring tool to detect tumor fraction and copy number alterations (CNAs) in cfDNA and CTCs, while whole-exome sequencing (WES) of matched CTCs, cfDNA, and bone marrow biopsies from patients with MM would help to resolve the clonal relatedness and role of liquid biopsies in the genomic monitoring of patients with MM.

CTCs and cfDNA are fundamentally different forms of liquid biopsy resulting from different biological processes. Substantial progress has been made for sequencing of CTCs and cfDNA, and studies have shown that each on its own may uncover similar somatic single nucleotide variants (SSNVs) and somatic copy number alterations (SCNAs) as detected in matched tumor biopsies[3–5]. For instance, we recently established that WES of CTCs in patients with MM is feasible and reflects the clonal composition of matched tumor biopsies[3]. Similarly, WES of cfDNA has been applied to patients with other types of cancers[6,7], but has yet to be evaluated in comparison with CTCs. Recent studies suggest that CTCs and cfDNA may uncover similar SSNVs based on the analysis of subsets of genes[8,9]. If concordant genome-wide SSNVs and SCNAs could be derived from cfDNA and CTCs in patients with MM, it could be possible to use CTCs and cfDNA interchangeably for comprehensive profiling of MM. However, if both provide different yet complementary information regarding clonal heterogeneity, then performing studies on both fractions of liquid biopsy could possibly replace the need for bone marrow biopsies in future clinical applications. Further, the analysis of CTCs and cfDNA may uncover additional SSNVs or SCNAs that were missed in the tumor biopsy but reflect other tumor sites in the body.

Despite the technical advances for sequencing of liquid biopsies, patients exhibit variability in yield and tumor fraction of CTCs and cfDNA, and this affects the ability to detect tumor mutations in all patient specimens. For instance, to enable WES of a limited number of CTCs (1–10 per tube of blood, as in most solid cancers), we have to isolate, amplify, and sequence each on its own[10]. Yet, in MM, the number of CTCs per tube of blood can be one or two orders of magnitude higher, affording the possibility to sequence enriched pools of CTCs in bulk and without whole-genome amplification.

We recently established a cost-effective approach, ichorCNA[11], to estimate tumor fraction in cfDNA using ULP-WGS (~0.1× coverage) and without prior knowledge of tumor mutations. We reasoned that the same approach could be applied to CD138-selected pools of MM-derived CTCs and cfDNA, to identify those with sufficient tumor fraction (≥5–10%) for WES[12]. We hypothesized that some patients may harbor higher tumor fraction in the CD138-selected CTCs than cfDNA or vice versa—due to technical or biological reasons—but that analyzing both may help to broaden the applicability of WES to patients with MM, particularly if CTCs and cfDNA exhibit concordant genomic profiles.

Here, we apply ULP-WGS and ichorCNA to estimate the tumor fraction in cfDNA and CD138-selected pools of CTCs from 107 and 56 patients with MM, respectively. We find that analyzing both CTCs and cfDNA leads to a higher fraction of patients having at least one sample with sufficient tumor content (≥5–10%) for WES. We then perform WES for 24 samples including matched cfDNA and bone marrow biopsies from nine patients and matched cfDNA, CTCs, and bone marrow biopsies from an additional four patients. We find that cfDNA and CTCs exhibit high concordance of SSNVs and SCNAs, reflect the clonal composition of the matched tumor biopsy, and uncover additional mutations exclusive to either CTCs or cfDNA, allowing blood-based tumor profiling in a greater fraction of patients.

## Results

**Detectability of CTCs and cfDNA in MM.** We first sought to evaluate the prevalence of circulating tumor DNA (tumor-derived cfDNA) and enriched CTCs in patients with MM. Building upon our recent work[11], we implemented a scalable process for isolation and ULP-WGS of cfDNA and CTCs in patients with MM. We specifically added the CD138-selection step to enrich CTCs without flow cytometry sorting (Supplementary Fig. 1). This process involves using ichorCNA to detect SCNAs and estimate tumor fraction from ULP-WGS data, and enables informed selection of samples for WES based on tumor content. We applied our integrated workflow to 107 cfDNA samples and 56 CTC samples with different stages of disease progression. These included samples from monoclonal gammopathy of undetermined significance (MGUS) (n = 9 cfDNA samples / n = 11 CTC samples), smoldering multiple myeloma (SMM) (n = 28 / n = 6), MM (n = 26 / n = 14) to relapsed disease (n = 44 / n = 25). The clinical data of these patients are included in Supplementary Data 1. For CTCs, the tumor fraction corresponds to the fraction of clonal cells present within the normal mononuclear cells isolated during the CD138+ bead selection used to enrich for the tumor cells.

We first examined tumor fractions in matched samples of cfDNA and enriched CTCs obtained from the same blood draw from 28 patients with MM (Fig. 1a, b). Interestingly, beside one patient (MM_2205), we found a wide discrepancy in the tumor fraction obtained from cfDNA and enriched CTCs, including several patients with higher tumor fractions in cfDNA than enriched CTCs or vice versa. For instance, patient MM_2214 had a tumor fraction of 80% in the enriched CTCs but only 6.7% in cfDNA. Conversely, patient MM_2242 had a tumor fraction of 91% in cfDNA and only 4% in the enriched CTCs. Indeed, there was no correlation between the tumor fraction present in cfDNA or enriched CTCs in the 28 samples that were performed on the same liquid biopsy, Fig. 1b, (Pearson correlation 0.081, p = 0.680). When we use either cfDNA or CTC tumor fractions ≥10% (the % required for confidence in performing WES on the sample), we could detect tumor DNA in 35% of the samples, rescuing some of the samples that were not detected by CTC or cfDNA alone (Fig. 1b). Our data suggest that analyzing both cfDNA and CTCs may broaden the applicability of liquid biopsies to patients with MM, provided that cfDNA and CTCs yield similar genomic profiles.

**Clinical correlation of CTCs and cfDNA in MM.** To further confirm the variability in tumor fractions among patients and potential correlations with clinical stage, we examined all 107 samples from patients for which cfDNA was isolated and 56 samples from patients for which CTCs were collected. Among 70 cfDNA and 39 CTC samples of overt myeloma samples (newly diagnosed or relapsed), there were 76%, 41%, and 24% of cfDNA

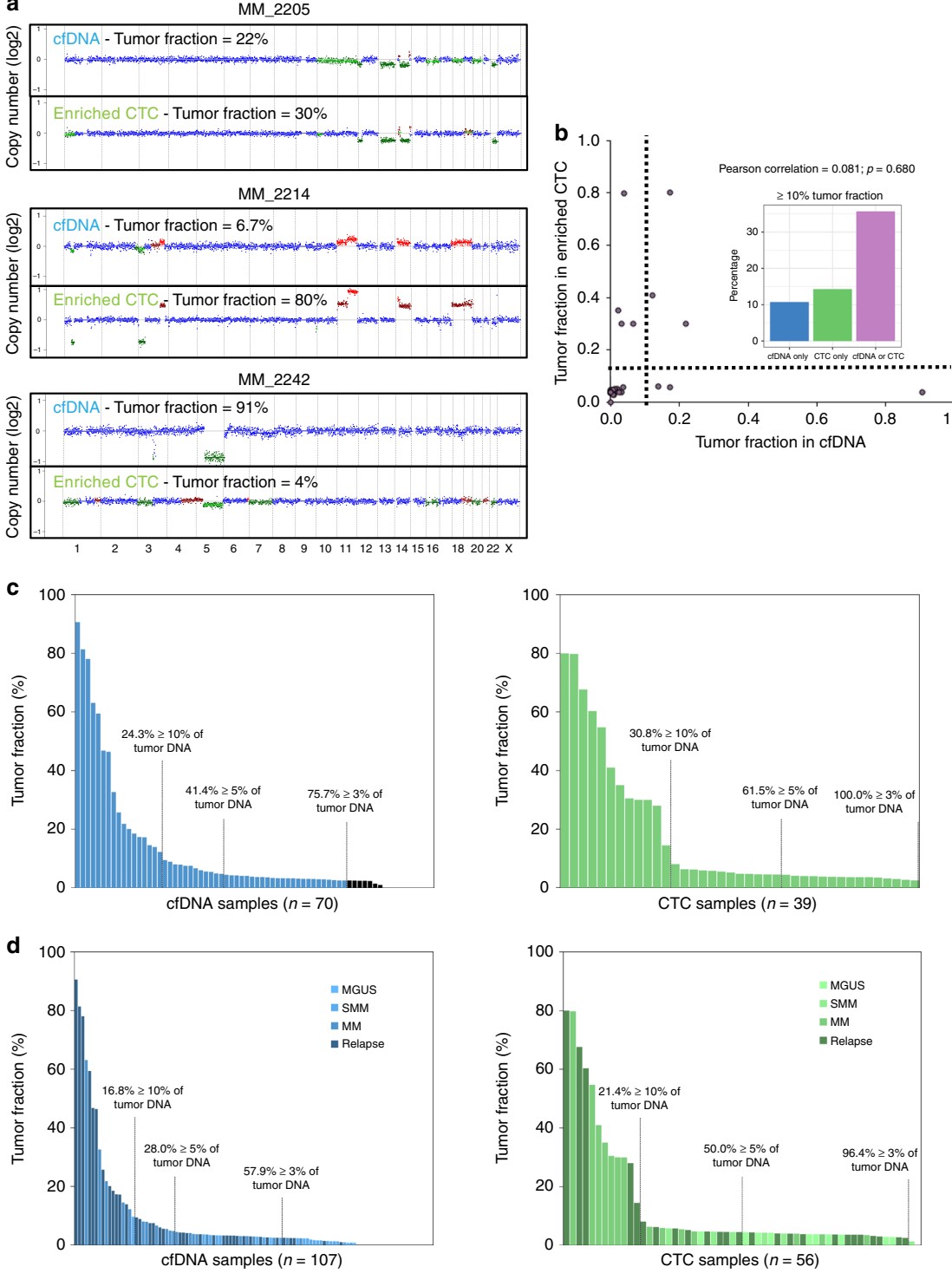

**Fig. 1** Detectability of CTCs and cfDNA in multiple myeloma. **a** Comparison of copy ratios between cfDNA ULP-WGS and CTC ULP-WGS from same MM patients. Amplification (red), deletion (green), and copy neutral (blue) are indicated. **b** Comparison of tumor fraction of matched cfDNA and CTC samples as determined via ULP-WGS and ichorCNA. Dashed line indicates 10% tumor fraction. Percentage of patients with ≥10% tumor fraction for cfDNA only (blue), CTC only (green), and cfDNA or CTC (purple) is indicated. Pearson correlation coefficient was calculated between the tumor fraction in cfDNA and CTC (Pearson correlation = 0.081; *p*-value = 0.680). **c** ULP-WGS tumor fraction estimates for 70 cfDNA samples and 39 enriched CTC samples from newly diagnosed MM and relapse patients; 76%, 41%, and 24% of cfDNA samples had ≥0.03, 0.05, and 0.1 tumor fractions, respectively (blue); 100%, 62%, and 31% of CTC samples had ≥0.03, 0.05, and 0.1 tumor fractions, respectively (green). **d** ULP-WGS tumor fraction estimates for 107 cfDNA samples and 56 CTC samples from MGUS, SMM, MM, and relapse patients; 58%, 28%, and 17% of cfDNA samples had ≥0.03, 0.05, and 0.1 tumor fractions, respectively (blue); 96%, 50%, and 21% of CTC samples had ≥0.03, 0.05, and 0.1 tumor fractions, respectively (green)

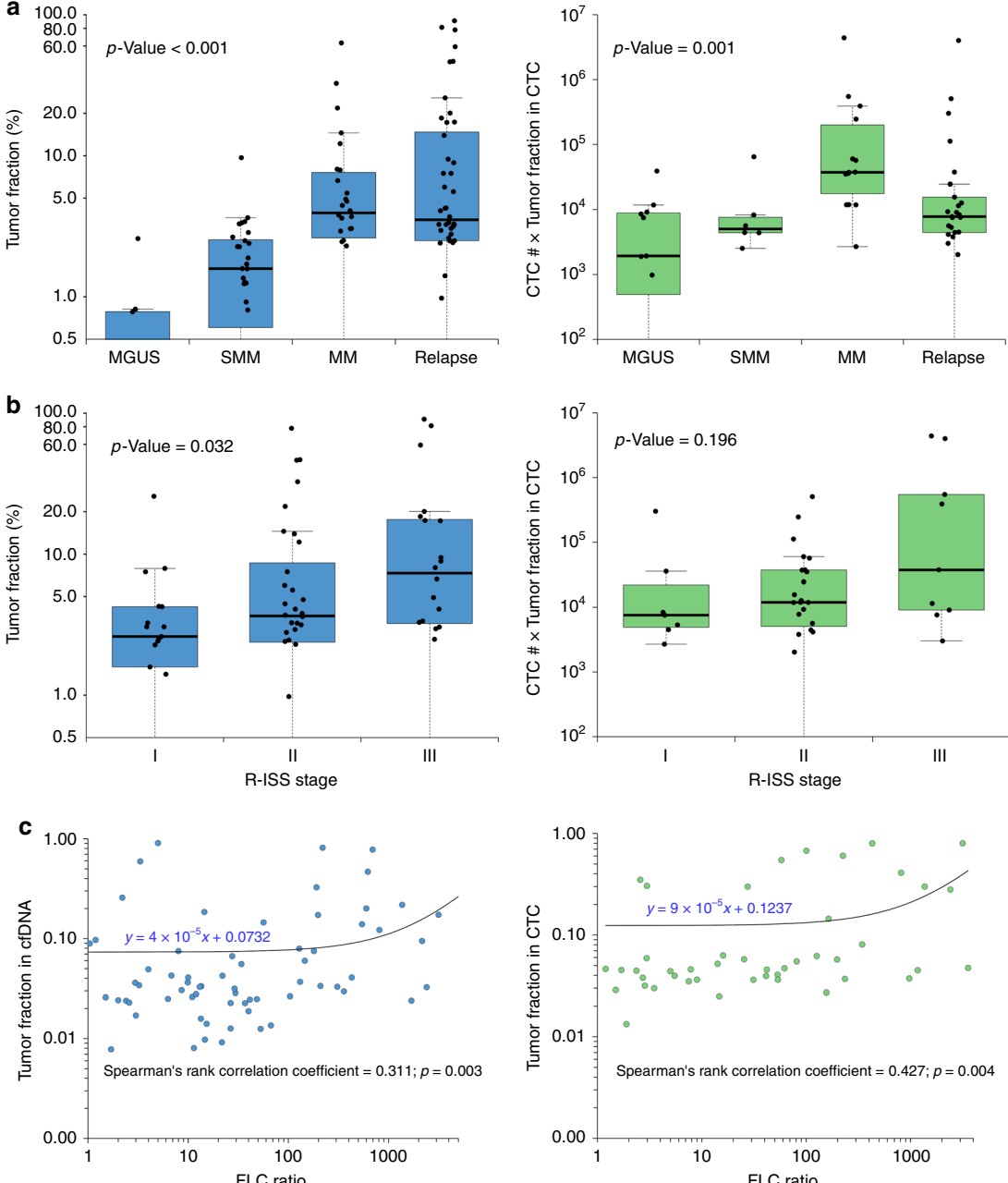

**Fig. 2** Clinical correlation of CTCs and cfDNA in multiple myeloma. The distributions of tumor fraction (%) in each group are shown as boxplots, where the central rectangle spans the first to the third quartile (interquartile range or IQR). A segment inside the rectangle shows the median, and whiskers above and below the box show the value 1.5 IQR above or below the third or the first quartile, respectively. **a** Tumor fractions of cfDNA samples (blue) and CTC number × tumor fractions of CTC samples (green) correlate with clinical stage of MM patients. The comparisons of the tumor fraction in cfDNA and CTC among different disease status (MGUS, SMM, newly diagnosed MM, and relapse) were performed by using Kruskal–Wallis test (p-value < 0.001 for cfDNA; p-value = 0.001 for CTC). **b** Tumor fractions of cfDNA samples (blue) correlate with ISS stage of MM patients, while CTC numbers × tumor fractions of CTC samples (green) do not. The comparisons of the tumor fraction in cfDNA and CTC among different R-ISS stages were performed by using Kruskal–Wallis test (p-value = 0.032 for cfDNA; p-value = 0.196 for CTC). **c** Tumor fractions of cfDNA (blue) and CTC counts (green: number of CD138-selected cells × tumor fractions in CD138-selected pools) in samples correlate with FLC ratio of MM patients. The correlations between the tumor fraction in cfDNA/CTC and serum free light chain ratio were analyzed by Spearman's rank correlation test (Spearman correlation coefficient = 0.311 and p-value = 0.003 for cfDNA; Spearman correlation coefficient = 0.427 and p-value = 0.004 for CTC)

samples with ≥3%, 5%, and 10% tumor fractions, respectively (Fig. 1c, blue panel). In comparison, there were 100%, 62%, and 31% of CTC samples having ≥3%, 5%, and 10% tumor fractions, respectively (Fig. 1c, green panel). Together, these data indicate that 76% and 100% of cfDNA and CTC samples, respectively, had

a tumor fraction above 3%, the lower limit of detection of ichorCNA as previously benchmarked[11]. When we include MGUS and SMM patients into cfDNA and CTC sample groups, 58%, 28%, and 17% of cfDNA samples had ≥3%, 5%, and 10% tumor fractions, and 96%, 50%, and 21% of enriched CTC

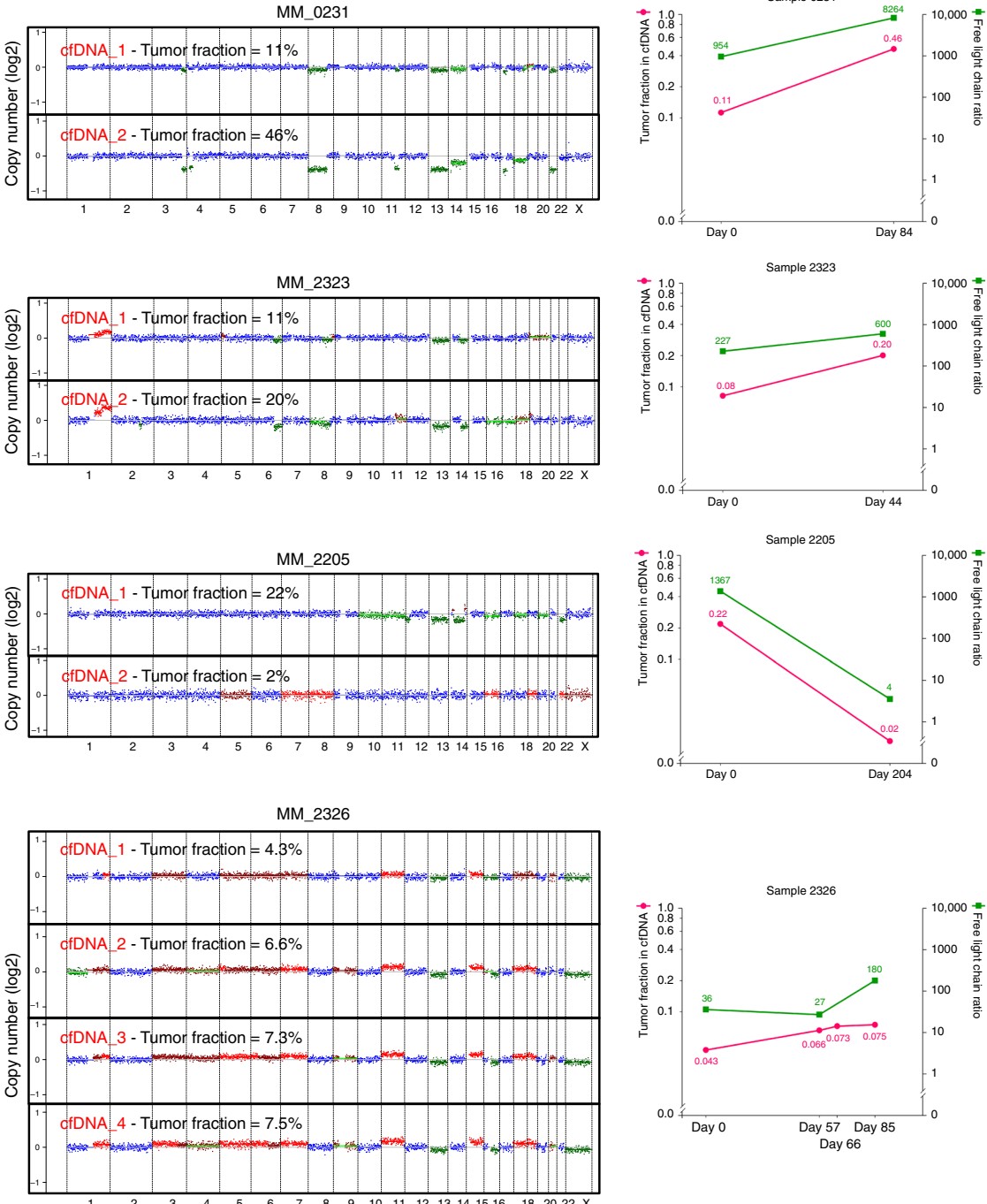

**Fig. 3** Monitoring progression or response to therapy with liquid biopsy. Comparison of copy ratios between cfDNA ULP-WGS from sequential samples from same MM patients. Tumor fraction (pink) and free light chain ratio (green) for each patient are also shown in this figure. Amplification (red), deletion (green), and copy neutral (blue) are indicated

samples had ≥3%, 5%, and 10% tumor fractions, respectively, suggesting that the tumor fraction increases with disease progression and that patients with MGUS and SMM have low tumor fractions given the earlier preclinical stages (Fig. 1d).

Interestingly, tumor fraction in cfDNA as well as CTCs (number of enriched CTC × tumor fraction) was significantly associated with the clinical stage of the disease (Fig. 2a—Kruskal–Wallis test, $p$-value <0.001), with a higher tumor fraction observed in cases with newly diagnosed MM compared to MGUS and SMM. The lower cfDNA and CTCs tumor fractions in

relapsed settings might be explained by the potentially lower tumor burden in early relapse patients. cfDNA also had a strong correlation with the revised international staging system (R-ISS)[13] and serum free light chain ratio (Fig. 2b, c, blue panel), while CTC only showed correlation with serum free light chain ratio but not with the R-ISS stage (Fig. 2b, c, green panel). Together, these results indicate that tumor fractions in liquid biopsy have a strong association with known biomarkers of MM, such as serum free light chain and R-ISS staging system.

**Clinical monitoring with cfDNA.** Moreover, to determine whether cfDNA can be used to track progression or response to therapy, we analyzed sequential cfDNA samples. As shown in Fig. 3, sample MM_0231 showed progression over a period of 2 months of follow-up while on the CD38 antibody Daratumumab therapy. There was an increase in tumor fraction from 11 to 46%, which was correlated with serum free light chain assay. A similar observation was seen in MM_2323. In contrast, patient MM_2205, who obtained a very good partial response to Carfilzomib, Revlimid, and Dexamethasone, had a decreased cfDNA tumor fraction from 22 to 2%. Finally, MM_2326 demonstrates that sequential samples are consistent and show reproducible data when there is stable disease burden in the patient and the samples are obtained weeks apart. Together, these four cases highlight the utility of cfDNA as a potential biomarker of disease response/progression in future studies via sequential samples during therapy, which is difficult to be performed using bone marrow biopsies.

**A comprehensive profile of clonal heterogeneity.** To investigate the concordance of the genomic profile of cfDNA and CTCs, we first examined large-scale (1 Mb) CNAs detectable by ULP-WGS. We focused on samples with ≥10% tumor content for SCNA analysis from ULP-WGS based on our prior benchmarking[11]. We identified 13q deletion in both cfDNA and CTC samples in MM_2205, 1p and 13q deletion as well as gain of 1q in MM_2213, and deletion 1p and gain of 11q in MM_2214 (Fig. 4a). In addition, there was a strong correlation in the large CNAs observed in matched cfDNA and CTC using ULP-WGS and tDNA using WES (Fig. 4b).

To next assess whether cfDNA or CTCs or both can capture the genetic diversity of MM, we performed WES on 24 samples including nine cases with matched cfDNA and tDNA and four cases with matched cfDNA, CTCs, and tDNA, along with germline control for all samples. The mean target coverage (MTC) was similar for all compartments (average, MTC = 204×). The mutational signatures were similar in all three compartments (Supplementary Fig. 2). Both clonal and subclonal SNVs were similarly dominated by C>T transitions at CpG sites. This mutational process has previously been associated with ageing or APOBEC[14]. In terms of clonal heterogeneity, 99% clonal mutations present in tDNA were confirmed in cfDNA or CTC (Fig. 5a, b—left Venn diagrams). Inversely, 94% of the mutations present in cfDNA or CTC were confirmed in tDNA (Fig. 5a, b—right Venn diagrams). CTC or cfDNA samples with higher tumor purity tended to uncover more mutations, expectedly, including a greater fraction of those present in the tumor biopsy. For patients with a similar tumor fraction in cfDNA and enriched CTCs (MM_2205 and MM_2017), sequencing of both CTCs and cfDNA still uncovered more mutations than either on its own.

To further explore the clonal heterogeneity among cfDNA, CTC, and tDNA, we performed cluster analysis for matched samples (Fig. 5c). We found the majority of mutations to reside in the shared clonal cluster for all cases. In most of the cases, subclonal clusters of mutations were identified in all three compartments; while in some cases, a subclone was detected in only one of the compartment (Fig. 5a, c). Of note, we observed several cases in which a cfDNA subclone was not detected in tDNA or CTC, and a CTC subclone was not detected in cfDNA or tumor biopsy. This indicates that either allelic fractions of these subclonal mutations were too low in the tumor biopsy sample to be detected or that the specific subclone is not present at the bone marrow biopsy site but only in a distant bone marrow or extra-medullary site. Most interestingly, the combination of CTCs and cfDNA was able to detect almost all clonal mutations

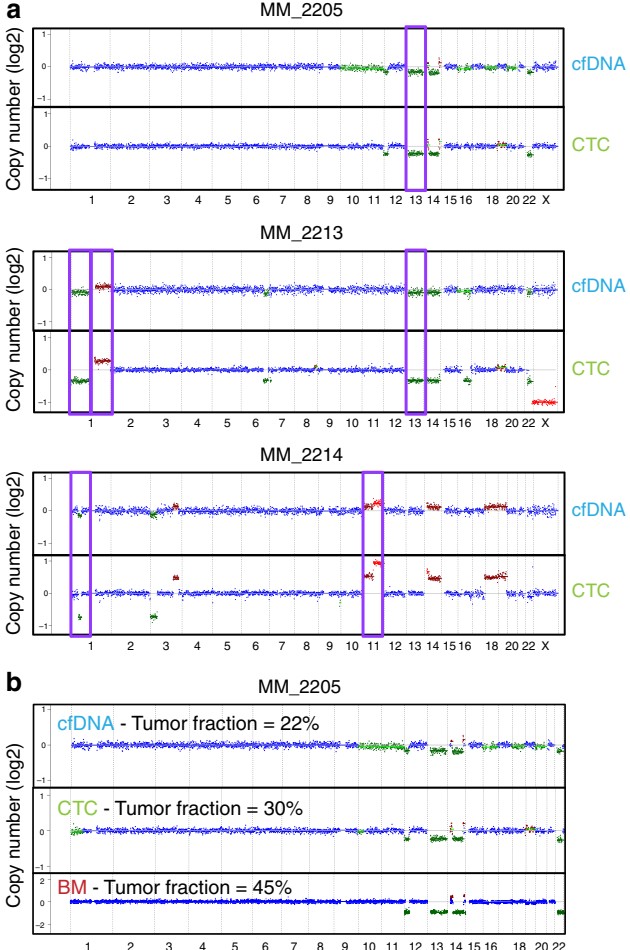

**Fig. 4** Copy number alterations detection of CTCs and cfDNA in multiple myeloma. **a** MM-related copy number alterations (CNAs—purple box) were detected in three patients with matched cfDNA and CTC samples via ULP-WGS. Amplification (red), deletion (green), and copy neutral (blue) are indicated. **b** Comparison of normalized copy ratios (log2) among cfDNA ULP-WGS, CTC ULP-WGS, and tumor biopsy WES in MM_2205. Amplification (red), deletion (green), and copy neutral (blue) are indicated

identified in the bone marrow biopsy sample and defined other subclones that were not identified in the bone marrow. For instance, a *TP53* subclone was only detected in cfDNA and CTCs in sample MM_2017 (Fig. 5a, c and Fig. 6a, c). Observation in both cfDNA and CTCs provides cross-validation for this mutation in two orthogonal sources of tumor DNA from the same tube of blood. Overall, an average of 96% of non-silent clonal mutations that were present in cfDNA was confirmed in CTC, while 84% of non-silent clonal mutations that were present in CTC was confirmed in cfDNA. Similarly, we observed in nine matched samples of cfDNA and tDNA, an average of 83% of non-silent clonal mutations that were present in tDNA was confirmed in cfDNA (Fig. 6a, b), while 88% of non-silent clonal mutations that were present in cfDNA was confirmed in tDNA (Fig. 6a, b). Most recurrently mutated genes in MM, such as *KRAS*, *NRAS*, *BRAF*, and *TP53*, as well as pan-cancer mutations and focal SCNAs were identified in matched cfDNA, CTCs, and/or tDNA across patients (Fig. 7).

Our data indicate that sequencing of both cfDNA and CTCs could capture the mutational landscape of bone marrow tumors and provide a comprehensive profile of clonal heterogeneity in

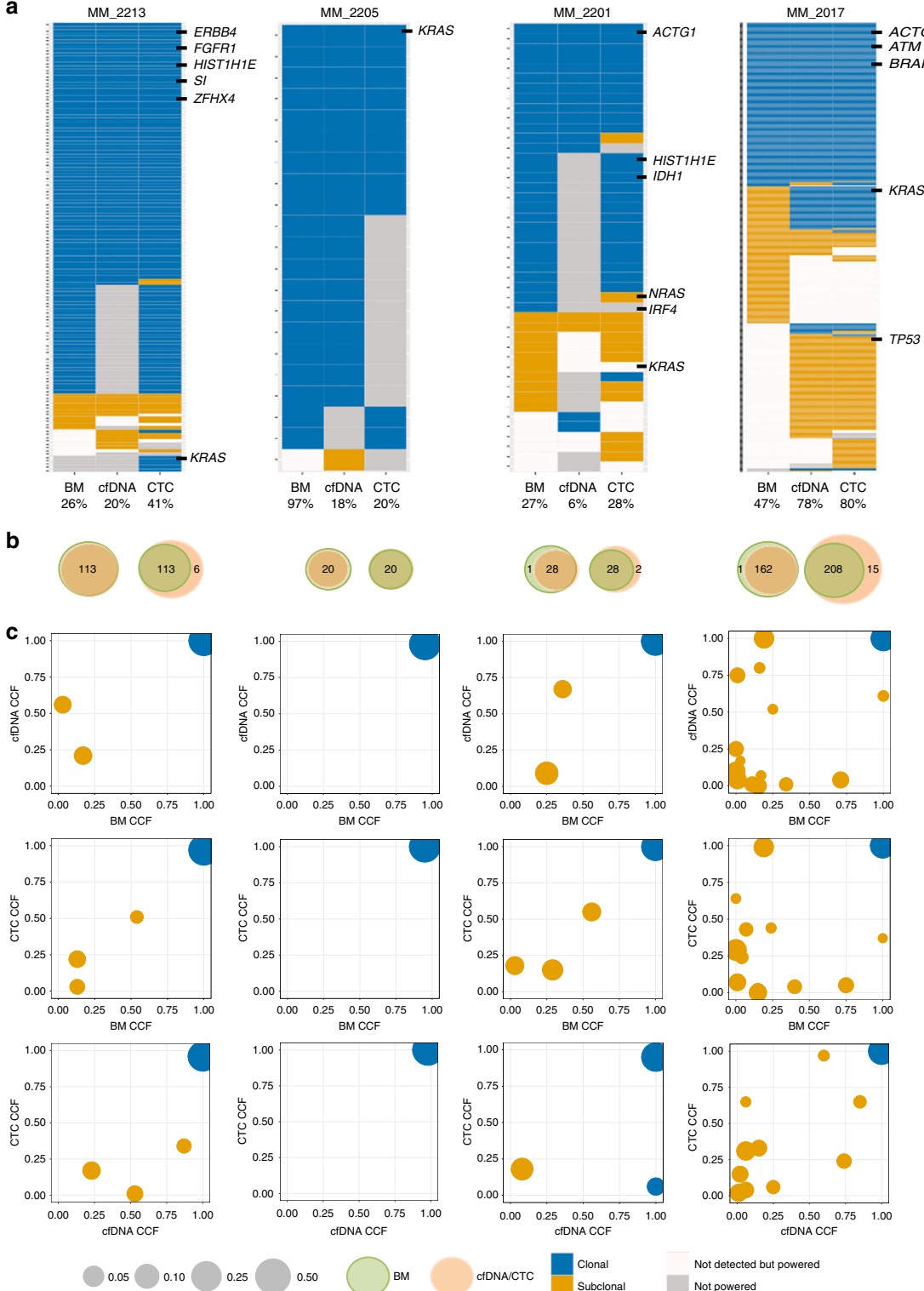

**Fig. 5** Whole-exome sequencing of matched cfDNA, CTCs, and BM tumor samples. **a** Presence of clonal (navyblue) and subclonal (yellow) somatic mutations in BM, cfDNA, and CTC WES is shown. Snow color represents mutations that were not detected with ≥0.9 detection power and gray color represents mutation sites with <0.9 detection power. MM and actionable pan-cancer related genes and purity of each sample are indicated. **b** Left Venn diagram shows the number of clonal mutations that were present in bone marrow biopsies (green) and confirmed in either cfDNA or CTC samples (orange). Right Venn diagram shows the number of clonal mutations that were present in either cfDNA or CTC samples (orange) and confirmed in bone marrow biopsies (green). **c** Cancer cell fraction (CCF) for clusters of SSNVs detected in bone marrow, cfDNA, and CTC samples from the same MM patient. Mutations were clustered by CCF for each pair of samples using PHYLOGIC. Clonal (navyblue) SSNVs were defined as events having ≥0.9 CCF in both samples. Subclonal (yellow) SSNVs were defined as events having <0.9 CCF in samples. Size of circles indicated the fraction of SSNVs. Mutations having ≥0.9 detection power in both samples are shown and clusters with <3 mutations are excluded

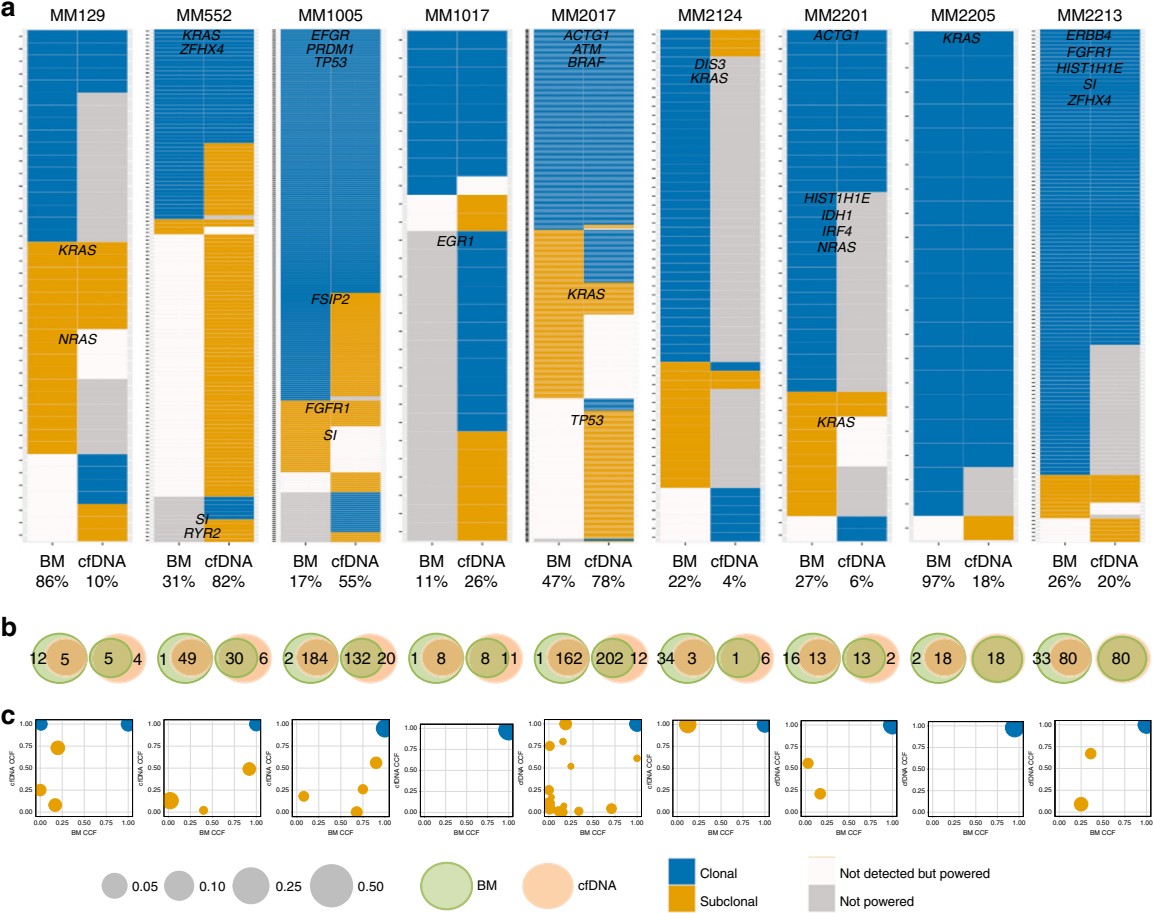

**Fig. 6** Whole-exome sequencing of matched cfDNA and BM tumor samples. **a** Presence of clonal (navyblue) and subclonal (yellow) somatic mutations in BM and cfDNA WES is presented. Mutations that were not detected with ≥0.9 detection power are shown in snow, and mutation sites with <0.9 detection power are shown in gray. MM and actionable pan-cancer related genes and purity of each sample are indicated. **b** Left Venn diagram shows the number of clonal mutations that were present in bone marrow biopsies (green) and confirmed in cfDNA samples (orange). Right Venn diagram shows the number of clonal mutations that were present in cfDNA samples (orange) and confirmed in bone marrow biopsies (green). **c** Cancer cell fraction (CCF) for clusters of SSNVs detected in bone marrow and cfDNA samples from the same MM patient. Mutations were clustered by CCF for each pair of samples using a PHYLOGIC. Clonal (navyblue) SSNVs were defined as events having ≥0.9 CCF in both samples. Subclonal (yellow) SSNVs were defined as events having <0.9 CCF in samples. Size of circles indicated the fraction of SSNVs. Mutations having ≥0.9 detection power in both samples are shown and clusters with <3 mutations are excluded

MM, enabling non-invasive profiling of tumor evolution using liquid biopsies.

## Discussion

CTCs and cfDNA are both important forms of liquid biopsy but are rarely analyzed together. Here, we performed a comprehensive comparison of enriched CTCs, cfDNA, and matched tumor biopsies in patients with MM. We find that patients exhibit variable yields of tumor-derived cfDNA and CTCs, but that collecting both cfDNA and CTCs increases the likelihood of isolating sufficient tumor DNA for WES. By applying WES, we find that CTCs and cfDNA exhibit high concordance in exome-wide SSNVs and SCNAs and uncover the majority of SSNVs and SCNAs present in the tumor biopsy. Our study suggests that CTCs and cfDNA may be used interchangeably for comprehensive genomic profiling of MM and, together, may broaden the applicability of liquid biopsies to patients with MM. These data are consistent with previous reports showing similarities in the mutational landscape of CTCs and cfDNA in metastatic breast cancers[15,16].

Our method, ichorCNA, enables simultaneous detection of SCNAs and quantification of tumor fraction in cfDNA using ULP-WGS[11]. In contrast to approaches such as deep profiling of recurrent SSNVs (17, 18) or VDJ rearrangements (19), ichorCNA provides a genome-wide assessment of tumor fraction using ULP-WGS and requires no prior knowledge of the mutations in a patient's tumor. Applying to 139 samples, we detected a tumor fraction of >0.10 in 17% and 21% of the cfDNA and CTC samples, respectively, sufficient for WES. We also detected >0.03 tumor fraction (the limit of detection for ichorCNA using ULP-WGS data) in 58% and 96% of the cfDNA and CTC samples, respectively. We therefore reasoned that tumor fraction as determined by ichorCNA may serve as a biomarker for patients with MM.

The correlation of the tumor fraction in cfDNA and enriched CTCs with disease progression from MGUS, SMM to overt MM as well as with biomarkers of disease progression and prognosis in MM indicate that the use of ULP-WGS can represent a novel biomarker for disease progression and response to therapy in MM. This will require further

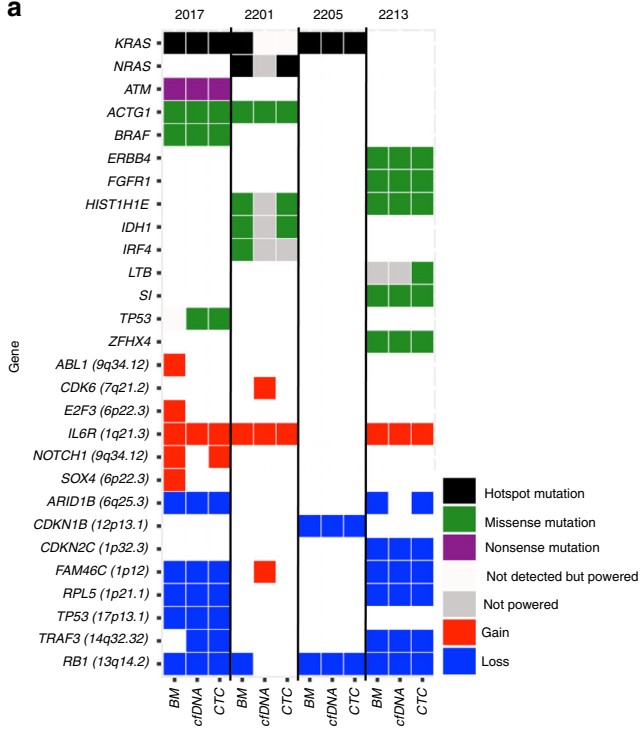

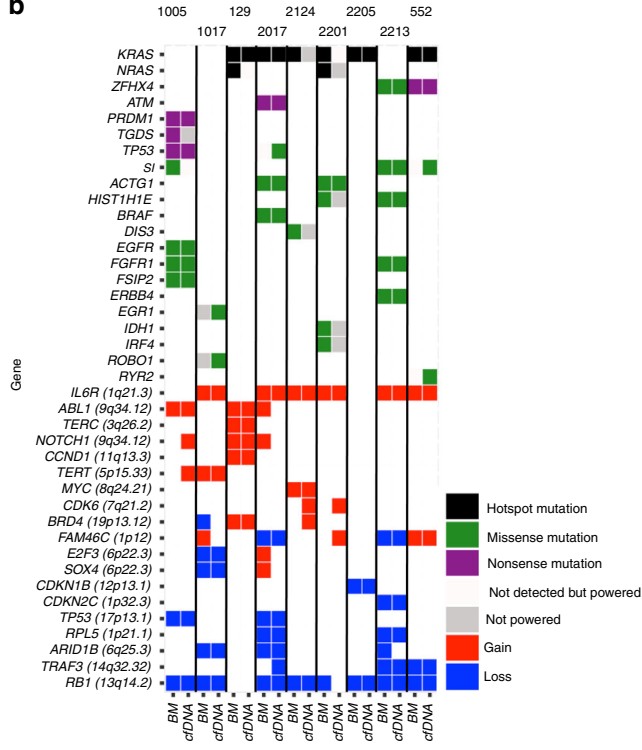

**Fig. 7** Somatic mutations and copy number alterations in matched cfDNA, CTCs, and BM tumor samples. **a** The alteration status of MM and actionable pan-cancer mutations and focal SCNAs are shown for bone marrow biopsies, cfDNA, and CTC samples from same MM patients. Hotspot mutation (black), missense mutation (green), nonsense mutation (purple), gain (red), and loss (blue) are indicated for specific SSNVs and SCNAs. Mutations that were not detected with ≥0.9 detection power (snow) and mutations with <0.9 detection power (gray) are distinguished in this figure. **b** The alteration status of MM and actionable pan-cancer mutations and focal SCNAs are shown for bone marrow biopsies and cfDNA samples from same MM patients. Hotspot mutation (black), missense mutation (green), nonsense mutation (purple), gain (red), and loss (blue) were indicated for specific SSNVs and SCNAs. Mutations that were not detected with ≥0.9 detection power (snow) and mutations with <0.9 detection power (gray) are distinguished in this figure. Mutations were predicted using Mutect and SCNAs were predicted using ReCapSeg

complexity with few mutations shared across many patients. Here, we demonstrate that the use of ULP-WGS along with WES would provide a more comprehensive approach to identify CNAs and somatic mutations in the peripheral blood of MM patients. Moreover, we among others have shown recently the feasibility of performing targeted sequencing on single cells of CTCs or WES on CTCs isolated by multi-color flow cytometry[3,10]. Both of these methods are interesting for research applications but would have limitations to be taken for clinical applications in larger number of samples. Therefore, we designed the CD138-selection step as a quick and easy way to enrich CTCs without the need for specialized equipment such as single cell isolation of flow cytometry sorting. The enrichment step can be performed in most clinical labs, but may result in substantial carryover of white blood cells and affect our ability to detect limited numbers of CTCs. The use of ULP-WGS can therefore be an essential tool to further define which samples should be used for further genomic evaluation. Indeed, the use of ULP-WGS for CNAs and possibly in future for the detection of translocations in the peripheral blood in MM would represent an important clinical tool.

Moreover, we demonstrated that sequential samples can help track in the future clonal evolution in patients with MM. Disease response or progression can be tracked with ULP-WGS and it was validated with bonafide biomarkers of disease progression. We envision that ULP-WGS can be used in sequential samples during therapy to define response and progression but more importantly, those samples can be further analyzed for the specific resistant clones that emerge during therapeutic interventions in these patients. This approach will require novel technologies with deep-targeted sequencing and potentially be used for detection of minimal residual disease.

Our data strongly suggest that the detection of CNAs and SSNVs in both CTCs and cfDNA can be complementary in replacing the use of bone marrow biopsies to track clonal heterogeneity in MM. Indeed, this liquid biopsy test can be more sensitive in detecting clones and subclones that were not identified in the bone marrow sample, potentially due to the limitation of sampling site of the bone marrow. Analyzing CTCs and cfDNA together enables cross-validation of mutations, uncovers additional mutations exclusive to either CTCs or cfDNA, and allows blood-based tumor profiling in a greater fraction of patients. Future studies using larger and prospective cohorts will help define the role of this approach in clinical practice.

## Methods

**Patient cohort**. A total of 139 patient samples were used for this study. All patients had active MM according to the IMWG criteria[17], except for 14 patients with monoclonal gammopathy of undetermined significance (MGUS) and

validation in larger prospective studies to confirm it as a possible biomarker in MM.

Interestingly, a recent study attempted to use a limited panel of oncogenic mutations in MM, and indeed showed that the detection of these mutations correlates with disease progression and response[4]. However, the study was limited to a few genes being tested and given that MM is characterized by genomic

33 patients with SMM. The review boards of participating centers approved the study, which was conducted according to the Declaration of Helsinki and International
Conference on Harmonization Guidelines for Good Clinical Practice. All patients provided written informed consent to allow the collection of blood and bone marrow analysis of clinical and genetic data for research purposes (IRB 07–150 and 14–174).

**DNA extraction**. Plasma samples were isolated from whole blood EDTA tubes after two-step centrifugation: 300×g for 10 min and 3000×g for 10 min. DNA was extracted using Qiagen circulating nucleic acid kits from 2 to 6 mL of plasma. CTCs and bone marrow plasma cells were isolated using CD138 bead selection after Ficoll of whole blood and bone marrow samples, respectively. Peripheral blood mononuclear cell (PBMC) negative fractions were used for germline DNA. Genomic DNA was extracted using Qiagen DNA extraction kit.

**cfDNA sequencing**. For ULP-WGS, libraries were prepared using the Kapa Hyper Prep kit with custom adapters (IDT and Broad Institute) starting with 5 ng of DNA. Up to 96 libraries were pooled and sequenced using 100 bp paired-end runs over 1× lane on a HiSeq2500 (Illumina). For WES, libraries were prepared using the Kapa Hyper Prep kit with custom adapters (IDT and Broad Institute) starting with 20 ng of DNA. Libraries were then quantified using the PicoGreen (Life Technologies) and pooled up to 12-plex. Hybrid capture of cfDNA libraries was performed using the Nextera Rapid Capture Exome kit (Illumina) with custom blocking oligos (IDT and Broad Institute). Sequencing was performed using 100 bp paired-end runs on Illumina HiSeq4000 in high-output mode with two to four libraries per lane.

**Genomic DNA sequencing**. Libraries were prepared and hybrid captured using the Nextera Rapid Capture Exome kit (Illumina) with 25 ng of DNA input. Sequencing was performed on Illumina HiSeq4000 in high-output mode with 100 bp paired-end reads. Two to four libraries were pooled per lane.

**Computational analyses**. Sequencing data were analyzed using the pipelines of the Broad Institute of Harvard and MIT (Firehose, www.broadinstitute.org/cancer/cga).
In order to estimate the quality and presence of tumor, we performed ULP-WGS of cfDNA and CTC to an average genome-wide fold coverage of 0.1×[11]. We analyzed the depth of coverage in a ULP sample to estimate large-scale CNAs and estimate the fraction of tumor in ULP-WGS, using ichorCNA[11,18,19]. Briefly, the genome was divided into $T$ non-overlapping windows, or bins, of 1 Mb. Aligned reads were counted based on overlap within each bin using the tools in HMMcopy Suite (http://compbio.bccrc.ca/software/hmmcopy/). The read counts were then normalized to correct for GC content and mappability biases, and then CNAs and tumor fraction were estimated using HMMcopy R package[18]. Low coverage samples (<0.05×) and low tumor fraction samples were manually reviewed for tumor fraction estimation.
The WES output was analyzed by the Broad Picard pipeline, resulting in BAM files aligned to hg19 with calibrated quality scores[20,21]. We used MuTect[12] within the Firehose framework to call somatic mutations in tumor biopsies, cfDNA, and CTC samples[20,22]. We assessed cross-sample contamination levels using ContEst[23] and filtered out potential artifactual OxoG mutations using the OxoG3 filter[24] and annotated mutation with Oncotater. Then, we realigned reads around mutated sites with Novoalign to hg19 including decoy sequences and re-ran MuTect to filter out mutations in problematic regions. To call somatic insertions and deletions (indels), we used Strelka[25] and annotated the mutation consequences using Oncotator. We also filtered out SSNVs and indels present in a panel of normal samples in order to filter out potential germline sites or recurrent artifactual sites. An additional filter for cfDNA and CTC samples was applied[11]. We applied a threshold of $LOD_T >$ 11.72 for C>A mutations at reference C sites that was previously identified[11] for our cfDNA samples.
To evaluate SSNVs in matched samples (Tumor biopsy, cfDNA, and/or CTC), we considered the union of mutations called in these samples, by forced calling to quantify the number of alternate reads at each mutation site[11]. To estimate somatic copy number alteration, we used ReCapSeg (http://gatkforums.broadinstitute.org/categories/recapseg-documentation), which calculated proportional coverage for each target region and then normalized each segment using the median proportional coverage in a panel of normal (PON) samples sequenced with the same capture technology. The sample was projected to a hyperplane defined by the PON and the tumor copy ratio was estimated. These copy-ratio profiles were segmented with CBS[26]. To estimate allelic copy number, germline heterozygous sites in the normal sample were called via GATK Haplotype Caller[21,27]. Then, the contribution of each homologous chromosome was assessed via reference and alternate read counts at the germline heterozygous sites. Finally, we segmented the allele specific copy ratios using PSCBS[26]. Resulting copy ratios and the force called SSNVs and indels were used as input for ABSOLUTE[28,29], to estimate the sample purity, ploidy, and cancer cell fraction (CCF) of SCNAs and SSNVs. ABSOLUTE solutions were manually reviewed and selected purity/ploidy solutions. As we expected that cfDNA and CTC would be derived from tumor cells related to tumor biopsies, we expected that the ploidy of tumor biopsies and cfDNA/CTC samples

would be consistent. To assess mutation clonality in matched samples, we used PHYLOGIC[30] to perform clustering of ABSOLUTE CCFs.
In order to compare with tumor biopsy, cfDNA, and CTC, we used the predicted ABSOLUTE CCF to assign clonal (≥0.9 CCF) and subclonal (<0.9 CCF) in the tumor biopsy. If there were ≥3 reads supporting the mutant allele, then considered confirmed in cfDNA or CTC. If the site had <0.9 power, then we considered the mutation unpowered. MM_2205 CTC sample used 0.7 power due to lower coverage. If a locus had <3 reads of the mutation allele in cfDNA/CTC and if the site had ≥0.9 power, then the mutation was powered but not confirmed in cfDNA/CTC.

**Statistical analysis**. Pearson correlation coefficient was calculated between the tumor fraction in cfDNA and CTC. The correlations between the tumor fraction in cfDNA/CTC and serum free light chain ratio were analyzed by Spearman's rank correlation test. The comparisons of the tumor fraction in cfDNA and CTC among different disease status (MGUS, SMM, newly diagnosed MM, and relapse) and R-ISS stages were performed by Kruskal–Wallis test. All statistical analyses were performed in R (version 3.1.1) and SAS 9.2 software (SAS Institute Inc., Cary, North Carolina). A $p$-value <0.05 was considered statistically significant.

**Data availability**. Sequencing data can be accessed via dbGaP accession code phs001323. All other remaining data are available within the Article and Supplementary Files, or available from the authors upon request.

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

## Acknowledgements

We would like to first and foremost acknowledge the courageous patients and their families for participation and contribution to this study. We are also grateful for all the members of Van Allen lab for their helpful discussions and feedback. We thank Tiana Issa for her graphic support. This work was supported in part by NCI R01 CA181683-01A1, the Multiple Myeloma Research Foundation (MMRF), and the Leukemia & Lymphoma Society (LLS). We would also like to acknowledge the generous support of the Gerstner Family Foundation.

## Author contributions

Designed studies: S.M., J.P., S.F, V.A., I.M.G.; performed experiments: S.M., M.C., M.B., S.R., C.J.L., G.H., D.H., G.G., K.Z.S., D.R., C.F., J.R., A.Y., A.P.G.; analysis: S.M., J.P., S.F., M.C., M.B., S.R., C.J.L., L.G., E.M.V.A., S.K., J.C.L., G.H., G.G., V.A., I.M.G.; writing—review and editing: all authors; funding acquisition: I.M.G.; M.C., M.B., S.F. and G.H. contributed equally to this work.

## Additional information

**Competing interests:** The authors declare no competing interests.

