## [Peer Review File · Nature Communications]

Reviewers' comments:

Reviewer #1 (Remarks to the Author):

The paper by Lohr et al should be included. Reference 4 discusses single cell analysis of prostate cancer. The appropriate reference should be.

<https://www.ncbi.nlm.nih.gov/pubmed/27807282>

It would be useful to include the following references.

<https://www.ncbi.nlm.nih.gov/pubmed/28385781>

<https://www.ncbi.nlm.nih.gov/pubmed/27899805>

<https://www.ncbi.nlm.nih.gov/pubmed/28183851>

Indeed, the above reference discusses Monitoring multiple myeloma by next-generation sequencing of V(D)J rearrangements from circulating myeloma cells and cell-free myeloma DNA.

Aside from more patients, it would be useful to discuss how the current paper is different from the above manuscript.

There should be a discussion as to how this technology differs from the reports above in terms of mutational analysis?

It would be helpful to provide a models of how MM mutations change over time as opposed to just showing figures 5,6,7 which while fine for a basic scientist, are more difficult to interpret for a clinician.

Reviewer #2 (Remarks to the Author):

The authors address an important topic in cancer research, i.e., the suitability of "liquid biopsies" in cancer patients. They have applied their newly developed method called ichorDNA (MS accepted in Nat. Communic.) to screen cfDNA and CTCs in blood samples from patients with multiple myeloma (MM). In a subset analysis, they compared the genomic results of cfDNA, CTCs and bone marrow tumor DNA obtained from a selected set of patients. The authors conclude that the analysis of ctDNA and CTCs in MM provides complementary results, which is an important take home message.

Comments

1) The study is well performed and the subset analyses on the 4-9 cases with matched cfDNA, CTCs and Tumor DNA from bone marrow provide interesting proof-of-principle results. However, this study demonstrates not "clinical utility", as indicated in the Abstract and the text (e.g., page 7, 1st paragr.). This term has been clearly defined in the biomarker field and the current study results - although interesting - will not change the treatment of MM patients based on the blood results yet.

2) The authors may also discuss the reports in solid cancers such as the work of Michael Speicher and his team in breast cancer (e.g., Breast Cancer Res. 2014), also demonstrating that ctDNA and CTC analyses are complementary. A recent discussion of this issue has been published by Alix-Panabieres & Pantel in Cancer Discovery 2016.

3) On page 6, it is stated that the tumor fraction of cfDNA and CTCs was significantly associated with the clinical stage of disease. However, Figure 2 also shows equal or lower levels in relapsed patients, which is surprising and deserves a comment.

4) The authors discuss biological reasons on pages 7/8 for the discrepant findings of ctDNA vs. CTCs. Are there any technical reasons (e.g., blood volume analysed, etc.) that may also explain the differences? What is the "success rate" and pre-analytical variables of the applied technologies?

5) What are the next steps to bring this technology into the clinic? Which interventional studies can be initiated to prove clinical utility?

Reviewer #3 (Remarks to the Author):

“Whole-Exome Sequencing of Cell-Free DNA and Circulating Tumor Cells in Multiple Myeloma” by Ghobrial et al proposes a combinatory analysis of cell-free DNA (cfDNA) and enriched circulating tumor cells (eCTC) for multiple myeloma (MM) for a more detailed exploration of the mutational and clonal landscape of a patient. For that the authors collected blood and bone marrow samples from 139 consenting patients at different stages of MM. Plasma samples were isolated from whole blood for cfDNA isolation (107 samples). CTCs and bone marrow biopsies were first enriched with CD138+ and then were isolated via magnetic bead selection. eCTCs were collected from 56 patients and bone marrow DNA was collected from 10 patients.

Utilizing ultra-low pass whole genome sequencing (ULP-WGS) with an approach called ichorCNA (Adalsteinsson et al, Nature Communications, accepted), the tumor fractions of cfDNA and eCTC samples for MM were estimated, and samples with sufficient tumor fraction ($\geq 5-10\%$) in either cfDNA or eCTC were selected for further examination with whole exome sequencing (WES). An important finding is that for 28 patients with matched cfDNA and eCTCs obtained from the same blood draw, there was a significant discrepancy in the overall tumor fraction from cfDNA and eCTC.

Q1: The authors need to clarify in how many patients such a discrepancy was not present - I assume none, but it has to be specified.

For instance, in the sample MM_2214 (Supplementary Figure 1.A), cfDNA has a tumor fraction of 6.7% whereas eCTC has a tumor fraction of 80%. On the contrary, MM_2242 has cfDNA with 91% tumor fraction with the corresponding eCTC at only 4%. Another example is MM_2205, in which cfDNA and eCTC has 22% and 30% tumor fractions respectively, which does not agree at all with the previous two examples. This led to the authors concluding that there was absolutely no correlation between the tumor fraction presence between cfDNA and CTC for various reasons, leading them to the above $\geq 5-10\%$ selection criterion in either cfDNA OR eCTC. This allowed certain samples with very low tumor fractions in cfDNA or eCTC to be permitted for WES in the next step and allowed the detection of tumor DNA in more samples overall (35% of the samples).

Q2: I am wondering what would happen if tumor fraction was derived by using CTC twice, applied to two independent blood samples obtained from the same blood draw. (The same question applied to the use of cfDNA twice as well).

The authors have found that, after jointly analyzing WES from cfDNA, enriched CTC, and tDNA in samples with sufficient tumor fraction, there was a significant concordance in terms of somatic copy number alterations and somatic clonal mutations among cfDNA, eCTC, and tDNA. However, certain events were observed to be exclusive to either cfDNA or eCTC, which suggests that these two might also be used in a complementary fashion for blood-based tumor profiling.

I believe that one of the most important results of the paper is that several cfDNA subclones were not detected in tDNA or eCTC, and several eCTC subclones were not detected in cfDNA or tumor biopsy. There has been a lot of work on common types of cancer that either use only cfDNA with tumor biopsies or only eCTC with tumor biopsies. Based on these results, such studies could have been potentially missing certain subclones which might play an important role in the progression of the cancer.

Q3. Why are there 107 cfDNA, 56 eCTC, and 10 bone marrow samples in a cohort of 139 patients (am I missing something)? It makes sense that most patients would not be in a condition to provide bone marrow biopsies, but I am not sure I understand the reason why the number of cfDNA and eCTC samples are not equal.

Response to Reviewers

Reviewer #1 (Remarks to the Author):

Comment 1. The paper by Lohr et al should be included. Reference 4 discusses single cell analysis of prostate cancer. The appropriate reference should be. <https://www.ncbi.nlm.nih.gov/pubmed/27807282>

Answer 1. We appreciate the reviewer's comment and have updated our manuscript with appropriate references page 3 (reference #5):
Lohr JG, Kim S, Gould J, Knoechel B, Drier Y, Cotton MJ, *et al.* Genetic interrogation of circulating multiple myeloma cells at single-cell resolution. *Sci Transl Med* **2016**;8(363):363ra147 doi 10.1126/scitranslmed.aac7037.

Comment 2. It would be useful to include the following references.
<https://www.ncbi.nlm.nih.gov/pubmed/28385781>
<https://www.ncbi.nlm.nih.gov/pubmed/27899805>
<https://www.ncbi.nlm.nih.gov/pubmed/28183851>

Indeed, the above reference discusses Monitoring multiple myeloma by next-generation sequencing of V(D)J rearrangements from circulating myeloma cells and cell-free myeloma DNA. Aside from more patients, it would be useful to discuss how the current paper is different from the above manuscript. There should be a discussion as to how this technology differs from the reports above in terms of mutational analysis?

Answer 2. We thank reviewer for this comments. We have updated the discussion of our manuscript, p. 9, as following:

“Our method, ichorCNA, enables simultaneous detection of SCNAs and quantification of tumor fraction in cfDNA using ULP-WGS (Adalsteinsson et al Nat Commun 2017). In contrast to approaches such as deep profiling of recurrent SSNVs (17,18) or VDJ rearrangements (19), ichorCNA provides a genome-wide assessment of tumor fraction using ULP-WGS and requires no prior knowledge of the mutations in a patient's tumor. Applying it to 139 samples, we detected a tumor fraction of greater than 0.10 in 17% and 21% of the cfDNA and CTC samples, respectively, sufficient for WES. We also detected greater than 0.03 tumor fraction (the limit of detection for ichorCNA using ULP-WGS data) in 58% and 96% of the cfDNA and CTC samples, respectively. We therefore reasoned that tumor fraction as determined by ichorCNA may serve as a biomarker for patients with MM.”

These references were added to the manuscript:

17. Mithraprabhu S, Khong T, Ramachandran M, Chow A, Klarica D, Mai L, *et al.* Circulating tumour DNA analysis demonstrates spatial mutational heterogeneity that coincides with disease relapse in myeloma. *Leukemia* **2017**;31(8):1695-705 doi 10.1038/leu.2016.366.

18. Rustad EH, Coward E, Skytoen ER, Misund K, Holien T, Standal T, *et al.* Monitoring multiple myeloma by quantification of recurrent mutations in serum. *Haematologica* **2017**;102(7):1266-72 doi 10.3324/haematol.2016.160564.
19. Oberle A, Brandt A, Voigtländer M, Thiele B, Radloff J, Schulenkorf A, *et al.* Monitoring multiple myeloma by next-generation sequencing of V(D)J rearrangements from circulating myeloma cells and cell-free myeloma DNA. *Haematologica* **2017**;102(6):1105-11 doi 10.3324/haematol.2016.161414.

Comment 3.

It would be helpful to provide a model of how MM mutations change over time as opposed to just showing figures 5,6,7, which while fine for a basic scientist, are more difficult to interpret for a clinician.

Answer 3. Although we appreciate reviewer's comments, Figure 5, 6, and 7 do not show MM mutations change over time. They show the differences/similarities of MM mutations among BM, cfDNA, and CTC samples at the same time point. Since the focus of the present study was the comparison between BM, CTC and cfDNA mutations, we did not show MM mutations change over time. However, we evaluated the use of ULP-WGS tumor fractions as a marker of response over time, in comparison with serum free light chain level. The data are represented in Figure 3.

Reviewer #2 (Remarks to the Author):

The authors address an important topic in cancer research, i.e., the suitability of "liquid biopsies" in cancer patients. They have applied their newly developed method called ichorDNA (MS accepted in *Nat. Commun.*) to screen cfDNA and CTCs in blood samples from patients with multiple myeloma (MM). In a subset analysis, they compared the genomic results of cfDNA, CTCs and bone marrow tumor DNA obtained from a selected set of patients. The authors conclude that the analysis of cfDNA and CTCs in MM provides complementary results, which is an important take home message.

Comments 1. The study is well performed and the subset analyses on the 4-9 cases with matched cfDNA, CTCs and Tumor DNA from bone marrow provide interesting proof-of-principle results. However, this study demonstrates not "clinical utility", as indicated in the Abstract and the text (e.g., page 7, 1st paragr.). This term has been clearly defined in the biomarker field and the current study results - although interesting - will not change the treatment of MM patients based on the blood results yet.

Answer 1. We agree that our evaluation of tumor fraction as a biomarker requires further evaluation and have revised the text accordingly in pg. 2:

"Here, we sought to determine the clonal relatedness of CTCs, cfDNA, and tumor biopsies from patients with multiple myeloma (MM) and explore their potential value for clinical monitoring."

Comment 2. The authors may also discuss the reports in solid cancers such as the work of Michael Speicher and his team in breast cancer (e.g., Breast Cancer Res. 2014), also demonstrating that ctDNA and CTC analyses are complementary. A recent discussion of this issue has been published by Alix-Panabieres & Pantel in Cancer Discovery 2016.

Answer 2. We appreciate this important point. The discussion was modified p. 9 as indicated here:

“Our study suggests that CTCs and cfDNA may be used interchangeably for comprehensive genomic profiling of MM and together, may broaden the applicability of liquid biopsies to patients with MM. These data are consistent with a previous report showing similarities in the mutational landscape of CTCs and cfDNA in metastatic breast cancers (15,16).”

These references were added to the manuscript:

15. Heidary M, Auer M, Ulz P, Heitzer E, Petru E, Gasch C, *et al.* The dynamic range of circulating tumor DNA in metastatic breast cancer. Breast Cancer Res **2014**;16(4):421 doi 10.1186/s13058-014-0421-y.

16. Alix-Panabieres C, Pantel K. Clinical Applications of Circulating Tumor Cells and Circulating Tumor DNA as Liquid Biopsy. Cancer Discov **2016**;6(5):479-91 doi 10.1158/2159-8290.CD-15-1483.

Comment 3. On page 6, it is stated that the tumor fraction of cfDNA and CTCs was significantly associated with the clinical stage of disease. However, Figure 2 also shows equal or lower levels in relapsed patients, which is surprising and deserves a comment.

Answer 3. We agree with the reviewer’s comment that this is an important point to clarify. As mentioned, our data demonstrates a correlation between cfDNA tumor fractions (TF) and the clinical stage (MGUS vs. SMM vs. newly diagnosed MM) and R-ISS. This result is suggestive that cfDNA TF and CTCs are reflective of the tumor burden. The reason that relapsed disease has less tumor burden is because clinically, we do not wait for patients to have a significant tumor burden to be considered “relapsed” and require a new therapy. Any progression of 25% or more in the M spike indicates relapse. In most of these patients, their overall tumor burden is not higher than when they were originally diagnosed as they are usually relapsing from a complete remission, so it is not surprising that their TF is not higher.

The text was modified accordingly, p. 6:

“The lower cfDNA and CTCs tumor fractions in relapsed settings might be explained by the potentially lower tumor burden in early relapse patients.”

Comment 4. The authors discuss biological reasons on pages 7/8 for the discrepant findings of ctDNA vs. CTCs. Are there any technical reasons (e.g., blood volume analysed, etc.) that

may also explain the differences? What is the "success rate" and pre-analytical variables of the applied technologies?

Answer 4. A key technical reason for discrepant findings in cfDNA vs. CTCs would be a difference in power to detect mutations among samples. Two factors that affect power to detect mutations would be the unique depth of coverage and tumor fraction of the sequencing library derived from the specimen (Cibulskis et al Nat Biotechnol 2013). Although we cannot fully control the factors that affect the unique coverage attainable using WES (e.g. mass and quality of DNA used for library construction), we measure unique coverage at each locus and use it to compute the power to detect each mutation. We take into account the detection power as we compare mutations among CTCs and cfDNA (see 'Computational analyses' section of Methods). Our protocols for sequencing library construction are optimized for 5 ng of DNA input. For cfDNA, 90% of our samples had sufficient DNA to construct sequencing libraries; meanwhile for CTCs, 100% of the samples had sufficient DNA. For each sample sequenced using WES, we report the mean target coverage (204x in pg. 7) and tumor fraction in Fig. 5 and 6.

Comment 5. What are the next steps to bring this technology into the clinic? Which interventional studies can be initiated to prove clinical utility?

Answer 5. We believe that our study provides a rationale to further evaluate the applicability of cfDNA and CTC sequencing in clinical practice with larger cohorts. We are already applying this technology to a clinical trial of patients with myeloma to detect sequential changes in the clones in response to therapy and detect minimal residual disease and early relapse. This work is still ongoing. We believe that this technology can possibly define early relapse and identify clonal evolution and selection under therapeutic interventions. In addition, we believe that early detection of clonal evolution in MGUS/smoldering myeloma can help define patients who are likely going to progress to overt myeloma and therefore may require earlier therapeutic interventions. These studies are ongoing but may have significant clinical and translational utility.

Reviewer #3 (Remarks to the Author):

"Whole-Exome Sequencing of Cell-Free DNA and Circulating Tumor Cells in Multiple Myeloma" by Ghobrial et al proposes a combinatory analysis of cell-free DNA (cfDNA) and enriched circulating tumor cells (eCTC) for multiple myeloma (MM) for a more detailed exploration of the mutational and clonal landscape of a patient. For that the authors collected blood and bone marrow samples from 139 consenting patients at different stages of MM. Plasma samples were isolated from whole blood for cfDNA isolation (107 samples). CTCs and bone marrow biopsies were first enriched with CD138+ and then were isolated via magnetic bead selection. eCTCs were collected from 56 patients and bone marrow DNA was collected from 10 patients.

Utilizing ultra-low pass whole genome sequencing (ULP-WGS) with an approach called ichorCNA (Adalsteinsson et al, Nature Communications, accepted), the tumor fractions of

cfDNA and eCTC samples for MM were estimated, and samples with sufficient tumor fraction ($\geq 5-10\%$) in either cfDNA or eCTC were selected for further examination with whole exome sequencing (WES). An important finding is that for 28 patients with matched cfDNA and eCTCs obtained from the same blood draw, there was a significant discrepancy in the overall tumor fraction from cfDNA and eCTC.

Comment 1. The authors need to clarify in how many patients such a discrepancy was not present - I assume none, but it has to be specified.

For instance, in the sample MM_2214 (Supplementary Figure 1.A), cfDNA has a tumor fraction of 6.7% whereas eCTC has a tumor fraction of 80%. On the contrary, MM_2242 has cfDNA with 91% tumor fraction with the corresponding eCTC at only 4%. Another example is MM_2205, in which cfDNA and eCTC has 22% and 30% tumor fractions respectively, which does not agree at all with the previous two examples. This led to the authors concluding that there was absolutely no correlation between the tumor fraction presence between cfDNA and CTC for various reasons, leading them to the above $\geq 5-10\%$ selection criterion in either cfDNA OR eCTC. This allowed certain samples with very low tumor fractions in cfDNA or eCTC to be permitted for WES in the next step and allowed the detection of tumor DNA in more samples overall (35% of the samples).

Answer 1. We found one patient (MM_2205) who had similar tumor fractions in both cfDNA and CTC (22% and 30%), while all the other patients had higher tumor fractions in either cfDNA or CTC. We have revised the text to clarify that the discrepancy was not present in this one patient, MM_2205 in pg. 5:

“Interestingly, beside one patient (MM_2205), we found a wide discrepancy in the tumor fraction obtained from cfDNA and enriched CTCs, including several patients with higher tumor fractions in cfDNA than enriched CTCs or vice versa.”

This suggests the advantage of choosing the optimal material (i.e. cfDNA or CTC) to follow clinical progression of disease.

Comment 2. I am wondering what would happen if tumor fraction was derived by using CTC twice, applied to two independent blood samples obtained from the same blood draw. (The same question applied to the use of cfDNA twice as well).

The authors have found that, after jointly analyzing WES from cfDNA, enriched CTC, and tDNA in samples with sufficient tumor fraction, there was a significant concordance in terms of somatic copy number alterations and somatic clonal mutations among cfDNA, eCTC, and tDNA. However, certain events were observed to be exclusive to either cfDNA or eCTC, which suggests that these two might also be used in a complementary fashion for blood-based tumor profiling.

I believe that one of the most important results of the paper is that several cfDNA subclones were not detected in tDNA or eCTC, and several eCTC subclones were not detected in cfDNA or tumor biopsy. There has been a lot of work on common types of cancer that either use only cfDNA with tumor biopsies or only eCTC with tumor biopsies. Based on these results, such

studies could have been potentially missing certain subclones which might play an important role in the progression of the cancer.

Answer 2. We appreciate reviewer’s important point. We actually have shown in our accepted manuscript (Stover et al, JCO accepted) that, even though yields of cfDNA can vary, tumor fraction and copy number profiles are reproducible from independent blood draws using ichorCNA and ULP-WGS. Figure (a) shows the schematic of independently processed same-day blood draws. Figure (b) shows that total cfDNA yield and tumor fraction per patient. Finally, Figure (c) and (d) show the representative ichorCNA copy number plots from same-day blood draws for the two patients with the highest detected tumor fraction. This suggests the reproducibility of independent blood samples.

We agree with the reviewer’s comments here and this is why we suggest that cfDNA and CTC analyses are complementary, and there is a significant advantage of choosing the optimal materials (i.e. cfDNA or CTC) to follow clinical progression of disease from the initial ULP-WGS analysis.

Comment 3. Why are there 107 cfDNA, 56 eCTC, and 10 bone marrow samples in a cohort of 139 patients (am I missing something)? It makes sense that most patients would not be in a condition to provide bone marrow biopsies, but I am not sure I understand the reason why the number of cfDNA and eCTC samples are not equal.

Answer 3. Not all samples had matching cfDNA and CTC samples available and in many cases the availability of a bone marrow sample on the same day was very challenging. Indeed, our ability to obtain peripheral blood and marrow samples from the same patients and isolate enough DNA from both cfDNA and CTC samples of the same patient was very challenging. This is why we had some samples that were matched with cfDNA and bone marrow or others with cfDNA and CTCs and bone marrows while others with only cfDNA.

REVIEWERS' COMMENTS:

Reviewer #1 (Remarks to the Author):

The authors have done a reasonable job addressing the preliminary but very interesting potential of their findings.

Reviewer #1 (Remarks to the Author):

The authors have done a reasonable job addressing the preliminary but very interesting potential of their findings.

Please note that while reviewer 2 and 3 did not provide additional comments to the author, they separately conveyed that their previous concerns have now been adequately addressed.

We are grateful for their insightful comments from the reviewers to improve our manuscript and pleased that they found our work comprehensive.